# Patch-Fool: Are Vision Transformers Always Robust Against Adversarial Perturbations?

**Yonggan Fu**∗**, Shunyao Zhang**∗**, Shang Wu**∗**, Cheng Wan & Yingyan Lin**
Department of Electrical and Computer Engineering, Rice University
`{yf22, sz74, sw99, chwan, yingyan.lin}@rice.edu`

## Abstract

Vision transformers (ViTs) have recently set off a new wave in neural architecture design thanks to their record-breaking performance in various vision tasks. In parallel, to fulfill the goal of deploying ViTs into real-world vision applications, their robustness against potential malicious attacks has gained increasing attention. In particular, recent works show that ViTs are more robust against adversarial attacks as compared with convolutional neural networks (CNNs), and conjecture that this is because ViTs focus more on capturing global interactions among different input/feature patches, leading to their improved robustness to local perturbations imposed by adversarial attacks. In this work, we ask an intriguing question: "Under what kinds of perturbations do ViTs become more vulnerable learners compared to CNNs?" Driven by this question, we first conduct a comprehensive experiment regarding the robustness of both ViTs and CNNs under various existing adversarial attacks to understand the underlying reason favoring their robustness. Based on the drawn insights, we then propose a dedicated attack framework, dubbed Patch-Fool, that fools the self-attention mechanism by attacking its basic component (i.e., a single *patch*) with a series of attention-aware optimization techniques. Interestingly, our Patch-Fool framework shows **for the first time** that ViTs are not necessarily more robust than CNNs against adversarial perturbations. In particular, we find that ViTs are more vulnerable learners compared with CNNs against our Patch-Fool attack which is consistent across extensive experiments, and the observations from Sparse/Mild Patch-Fool, two variants of Patch-Fool, indicate an intriguing insight that *the perturbation density and strength on each patch seem to be the key factors that influence the robustness ranking between ViTs and CNNs*. It can be expected that our Patch-Fool framework will shed light on both future architecture designs and training schemes for robustifying ViTs towards their real-world deployment. Our codes are available at `https://github.com/RICE-EIC/Patch-Fool`.

## 1 Introduction

The recent performance breakthroughs achieved by vision transformers (ViTs) (Dosovitskiy et al., 2020) have fueled an increasing enthusiasm towards designing new ViT architectures for different vision tasks, including object detection (Carion et al., 2020; Beal et al., 2020), semantic segmentation (Strudel et al., 2021; Zheng et al., 2021; Wang et al., 2021), and video recognition (Arnab et al., 2021; Liu et al., 2021b; Li et al., 2021b; Fan et al., 2021). To fulfill the goal of deploying ViTs into real-world vision applications, the security concern of ViTs is of great importance and challenge, especially in the context of adversarial attacks (Goodfellow et al., 2014), under which an imperceptible perturbation onto the inputs can mislead the models to malfunction.

In response, the robustness of ViTs against adversarial attacks has attracted increasing attention. For example, recent works (Bhojanapalli et al., 2021; Aldahdooh et al., 2021; Shao et al., 2021) find that in addition to ViTs' decent task performances, they are more robust to adversarial attacks compared with convolutional neural networks (CNNs) under comparable model complexities. In particular, (Shao et al., 2021) claims that ViTs focus more on capturing the global interaction among

---

∗Equal contribution.

input/feature patches via its self-attention mechanism and the learned features contain less low-level information, leading to superior robustness to the local perturbations introduced by adversarial attacks. A natural response to this seemingly good news would be determining whether ViTs are truly robust against all kinds of adversarial perturbations or if their current win in robustness is an inevitable result of biased evaluations using existing attack methods that are mostly dedicated to CNNs. To unveil the potential vulnerability of ViTs, this work takes the first step in asking an intriguing question: "Under what kinds of perturbations do ViTs become more vulnerable learners compared to CNNs?", and makes the following contributions:

- We propose a new attack framework, dubbed Patch-Fool, aiming to fool the self-attention mechanism by attacking the basic component (i.e., a single patch) participating in ViTs' self-attention calculations. Our Patch-Fool attack features a novel objective formulation, which is then solved by Patch-Fool's integrated attention-aware patch selection technique and attention-aware loss design;

- We evaluate the robustness of both ViTs and CNNs against our Patch-Fool attack with extensive experiments and find that ViTs are consistently less robust than CNNs across various attack settings, indicating that ViTs are not always robust learners and their seeming robustness against existing attacks can be overturned under dedicated adversarial attacks;

- We further benchmark the robustness of both ViTs and CNNs under two variants of Patch-Fool, i.e., Sparse Patch-Fool and Mild Patch-Fool, and discover that the perturbation density, defined as the number of perturbed pixels per patch, and the perturbation strength highly influence the robustness ranking between ViTs and CNNs, where our Patch-Fool is an extreme case of high perturbation density and strength.

We believe our work has opened up a new perspective for exploring ViTs' vulnerability and understanding the different behaviors of CNNs and ViTs under adversarial attacks, and can provide insights to both future architecture designs and training schemes for robustifying ViTs towards their real-world deployment.

## 2 RELATED WORKS

**Vision transformers.** Motivated by the great success of Transformers in the natural language processing (NLP) field (Vaswani et al., 2017), ViTs have been developed by splitting an input image into a series of image patches and adopting self-attention modules for encoding the image (Dosovitskiy et al., 2020), and been shown to achieve competitive or superior performance over CNNs via dedicated data augmentation (Touvron et al., 2021) or self-attention structures (Yang et al., 2021; Graham et al., 2021; Liu et al., 2021a). As such, there has been tremendously increased attention on applying ViTs to various computer vision applications, such as self-supervised learning (Caron et al., 2021; Chen et al., 2021b; Xie et al., 2021; Li et al., 2021a), object detection (Carion et al., 2020; Beal et al., 2020), and semantic segmentation (Strudel et al., 2021; Zheng et al., 2021; Wang et al., 2021). The achievable performance of ViTs are continuously refreshed by emerging ViT variants, which provide new arts for designing ViT architectures. For example, convolutional modules have been incorporated into ViTs for capturing low-level features (Xiao et al., 2021; Wu et al., 2021; Graham et al., 2021; Peng et al., 2021), and replacing the global self-attention mechanism with local self-attention modules (Liu et al., 2021a; Dong et al., 2021; Liang et al., 2021; Liu et al., 2021b; Chu et al., 2021) has further pushed forward ViTs' achievable accuracy-efficiency trade-off. Motivated by the growing interest in deploying ViTs into real-world applications, this work aims to better understand the robustness of ViTs and to develop adversarial attacks dedicated to ViTs.

**Adversarial attack and defense.** Deep neural networks (DNNs) are known to be vulnerable to adversarial attacks (Goodfellow et al., 2014), i.e., imperceptible perturbations onto the inputs can mislead DNNs to make wrong predictions. As adversaries, stronger attacks are continuously developed, including both white-box (Madry et al., 2017; Croce & Hein, 2020; Carlini & Wagner, 2017; Papernot et al., 2016; Moosavi-Dezfooli et al., 2016) and black-box ones (Chen et al., 2017; Ilyas et al., 2018b; Andriushchenko et al., 2020; Guo et al., 2019; Ilyas et al., 2018a), which aggressively degrade the performances of the target DNN models. In particular, (Brown et al., 2017; Liu et al., 2020) build universal adversarial patches that are able to attack different scenes and (Liu et al., 2018; Zhao et al., 2020; Hoory et al., 2020) adopt adversarial patches to attack object detectors. However, these works focus on merely CNNs, questions regarding (1) whether patch-wise attacks

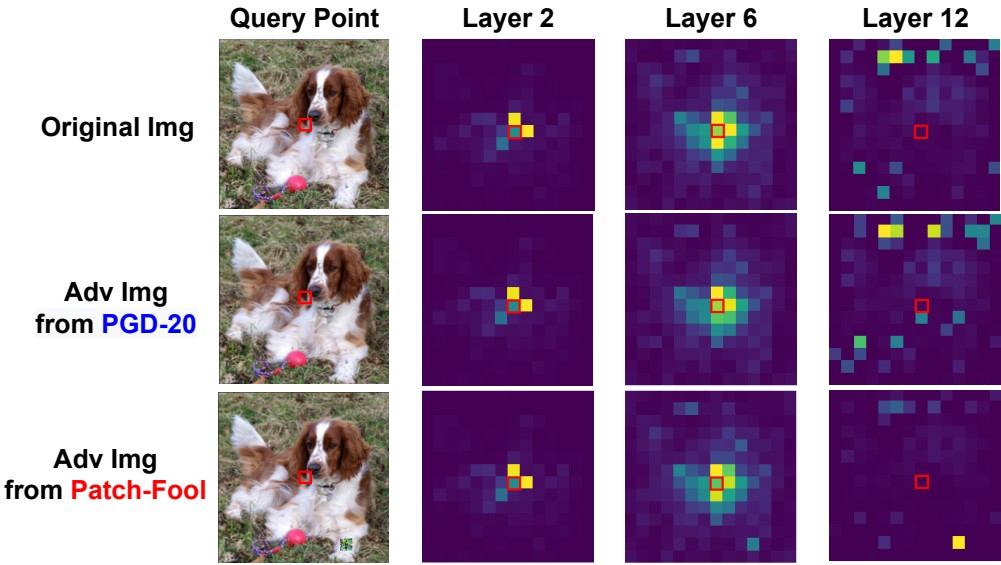

Figure 1: Comparisons among the attention maps in the intermediate layers of DeiT-S generated by the clean inputs, the adversarial inputs under PGD-20 attacks ($\epsilon = 0.003$), and the proposed Patch-Fool attack, respectively. In particular, we average the attention scores across all the attention heads in each layer and visualize the attention score of each token for a given query token (the center patch in the red box in our show case), following (Kim et al., 2021). We can observe that the difference in attention maps between clean and adversarial inputs generated by PGD-20 keeps small across different layers; In contrast, the proposed Patch-Fool notably enlarges the gap between clean and adversarial attention maps, demonstrating a successful attack for ViTs.

are effective for ViTs as compared to CNNs, and (2) how to efficiently construct strong patch-wise attacks utilizing the unique structures of ViTs are still under-explored yet interesting to be studied, especially considering patches are the basic elements for composing the inputs of ViTs. In response, various defense methods (Guo et al., 2017; Xie et al., 2017; Cohen et al., 2019; Metzen et al., 2017; Feinman et al., 2017; Fu et al., 2021a;b; Shafahi et al., 2019; Madry et al., 2017; Wong et al., 2019) have been proposed to improve DNNs' robustness against those attacks. The readers are referred to (Akhtar & Mian, 2018; Chakraborty et al., 2018) for more attack and defense methods.

**Robustness of vision transformers.** Driven by the impressive performance recently achieved by ViTs in various vision tasks, their robustness has gained increasing attention. A consistent observation drawn by pioneering works that study ViTs' robustness is that ViTs are more robust to adversarial attacks than CNNs since ViTs are more capable of capturing the global interactions among patches, while CNNs focus on local features and thus are more vulnerable to local adversarial perturbations. In particular, (Bhojanapalli et al., 2021) shows that ViT models pretrained with a sufficient amount of data are at least as robust as their ResNet counterparts on a broad range of perturbations, including natural corruptions, distribution shifts, and adversarial perturbations; (Aldahdooh et al., 2021) finds that vanilla ViTs or hybrid-ViTs are more robust than CNNs under $L_p$-based attacks; and (Shao et al., 2021) further explains that ViTs' learned features contain less low-level information and are more generalizable, leading to their superior robustness, and introducing convolutional blocks that extract more low-level features will reduce the ViTs' adversarial robustness. In addition, ViTs' adversarial transferability has also been studied: (Mahmood et al., 2021) shows that adversarial examples do not readily transfer between CNNs and transformers and (Naseer et al., 2021; Wei et al., 2021) propose techniques to boost the adversarial transferability between ViTs and from ViTs to CNNs. In parallel, (Mao et al., 2021) refines ViTs' architecture design to improve robustness. In our work, we challenge the common belief that ViTs are more robust than CNNs, which is concluded based on evaluations using existing attack methods, and propose to customize adaptive attacks utilizing ViTs' captured patch-wise global interactions to make ViTs weaker learners.

## 3 THE PROPOSED PATCH-FOOL FRAMEWORK

In this section, we present our Patch-Fool attack method that perturbs a whole patch to fool ViTs and unveils a vulnerable perspective of ViTs.

## 3.1 PATCH-FOOL: VALIDATING AND RETHINKING THE ROBUSTNESS OF ViTs

We extensively evaluate the robustness of several representative ViT variants against four state-of-the-art adversarial attacks (i.e., PGD (Madry et al., 2017), AutoAttack (Croce & Hein, 2020), CW-$L_\infty$ (Carlini & Wagner, 2017), and CW-$L_2$) with different perturbation strengths in Appendix. A.1. We observe that (1) ViTs are consistently more robust than CNNs with comparable model complexities under all attack methods, which is consistent with the previous observations (Bhojanapalli et al., 2021; Aldahdooh et al., 2021; Shao et al., 2021), and (2) ViT variants equipped with local self-attention ((Swin (Liu et al., 2021a))) or convolutional modules (LeViT (Graham et al., 2021)), which improve the model capability in capturing local features and thus boosts the clean accuracy, are more vulnerable to adversarial attacks, although they are still more robust than CNNs with comparable complexities. This indicates that the global attention mechanism itself can serve as a good robustification technique against existing adversarial attacks, even in lightweight ViTs with small model complexities. For example, as shown in Fig. 1, the gap between the attention maps generated by clean and adversarial inputs in deeper layers remains small. We are curious about "Are the global attentions in ViTs truly robust, or their vulnerability has not been fully explored and exploited?". To answer this, we propose our customized attack in the following sections.

## 3.2 PATCH-FOOL: MOTIVATION

Given the insensitivity of ViTs' self-attention mechanism to local perturbations, we pay a close attention to the basic component (i.e., a single patch) participating in the self-attention calculation, and hypothesize that customized adversarial perturbations onto a patch can be more effective in fooling the captured patch-wise global interactions of self-attention modules than attacking the CNN modules. This is also inspired by the word substitution attacks (Alzantot et al., 2018; Ren et al., 2019; Jin et al., 2019; Zang et al., 2019) to Transformers in NLP tasks, which replace a word with its synonyms, and here an image patch in ViTs serves a similar role as a word.

## 3.3 PATCH-FOOL: SETUP AND OBJECTIVE FORMULATION

**Attack setup.** In our proposed Patch-Fool Attack, we do not limit the perturbation strength onto each pixel and, instead, constrain all the perturbed pixels within one patch (or several patches), which can be viewed as a variant of sparse attacks (Dong et al., 2020; Modas et al., 2019; Croce & Hein, 2019). Such attack strategies will lead to adversarial examples with a noisy patch as shown in Fig. 1, which visually resembles and emulates natural corruptions in a small region of the original image, e.g., one noisy patch only counts for 1/196 in the inputs of DeiT-S (Touvron et al., 2021), caused by potential defects of the sensors or potential noises/damages of the optical devices.

**Objective formulation.** Given the loss function $J$ and a series of input image patches $\mathbf{X} = [\mathbf{x}_1, \cdots, \mathbf{x}_n]^\top \in \mathbb{R}^{n \times d}$ with its associated label $y$, the objective of our adversarial algorithm can be formulated as:

$$\underset{1 \leq p \leq n, \mathbf{E} \in \mathbb{R}^{n \times d}}{\arg \max} \quad J(\mathbf{X} + \mathbb{1}_p \odot \mathbf{E}, y) \tag{1}$$

where $\mathbf{E}$ denotes the adversarial perturbation, $\mathbb{1}_p \in \mathbb{R}^n$ such that $\mathbb{1}_p(i) = \begin{cases} 0, & i \neq p \\ 1, & i = p \end{cases}$ is a one hot vector, and $\odot$ represents the penetrating face product such that $\mathbf{a} \odot \mathbf{B} = [\mathbf{a} \circ \mathbf{b}_1, \cdots, \mathbf{a} \circ \mathbf{b}_d]$ where $\circ$ is the Hadamard product and $\mathbf{b}_j$ is the $j$-th column of matrix $\mathbf{B}$. For solving Eq. 1, our Patch-Fool needs to (1) select the adversarial patch $p$, and (2) optimize the corresponding $\mathbf{E}$ as elaborated in Sec. 3.4 and Sec. 3.5, respectively.

## 3.4 PATCH-FOOL: DETERMINE $p$ VIA ATTENTION-AWARE PATCH SELECTION

Denoting $\boldsymbol{a}^{(l,h,i)} = [a_1^{(l,h,i)}, \cdots, a_n^{(l,h,i)}] \in \mathbb{R}^n$ as the attention distribution for the $i$-th token of the $h$-th head in the $l$-th layer. For each layer $l$, we define:

$$s_j^{(l)} = \sum_{h,i} a_j^{(l,h,i)} \tag{2}$$

which measures the importance of the $j$-th token in the $l$-th layer based on its contributions to other tokens in the self-attention calculation. For better fooling ViTs, we select the most influential patch $p$ derived from $\arg \max_j s_j^{(l)}$ according to a predefined value $l$. We fix $l = 5$ by default since the patches

at later self-attention layers are observed to be diverse from the input patches due to the increased information mixed from other patches, making them non-ideal for guiding the selection of input patches as justified in Sec. 4.3.

### 3.5 PATCH-FOOL: OPTIMIZE E VIA ATTENTION-AWARE LOSS

Given the selected adversarial patch index $p$ from the above step, we define the *attention-aware loss* for the $l$-th layer as follows:

$$J_{\text{ATTN}}^{(l)}(\mathbf{X}, p) = \sum_{h,i} a_p^{(l,h,i)} \tag{3}$$

which is expected to be maximized so that the adversarial patch $p$, serving as the target adversarial patch, can attract more attention from other patches for more effectively fooling ViTs. The perturbation $\mathbf{E}$ is then updated based on both the final classification loss, i.e., the cross-entropy loss $J_{\text{CE}}$, and a layer-wise attention-aware loss:

$$J(\widetilde{\mathbf{X}}, y, p) = J_{\text{CE}}(\widetilde{\mathbf{X}}, y) + \alpha \sum_l J_{\text{ATTN}}^{(l)}(\widetilde{\mathbf{X}}, p) \tag{4}$$

where $\widetilde{\mathbf{X}} \triangleq \mathbf{X} + \mathbb{1}_p \odot \mathbf{E}$ and $\alpha$ is a weighted coefficient for controlling $\sum_l J_{\text{ATTN}}^{(l)}(\widetilde{\mathbf{X}}, p)$. We further adopt PCGrad (Yu et al., 2020) to avoid the gradient conflict of two losses, and thus the update of perturbation $\mathbf{E}$ is calculated using the following equation

$$\delta_{\mathbf{E}} = \nabla_{\mathbf{E}} J(\widetilde{\mathbf{X}}, y, p) - \alpha \sum_l \beta_l \nabla_{\mathbf{E}} J_{\text{CE}}(\widetilde{\mathbf{X}}, y) \tag{5}$$

where

$$\beta_l = \begin{cases} 0, & \left\langle \nabla_{\mathbf{E}} J_{\text{CE}}(\widetilde{\mathbf{X}}, y), \nabla_{\mathbf{E}} J_{\text{ATTN}}^{(l)}(\widetilde{\mathbf{X}}, p) \right\rangle > 0 \\ \frac{\left\langle \nabla_{\mathbf{E}} J_{\text{CE}}(\widetilde{\mathbf{X}}, y), \nabla_{\mathbf{E}} J_{\text{ATTN}}^{(l)}(\widetilde{\mathbf{X}}, p) \right\rangle}{\|\nabla_{\mathbf{E}} J_{\text{CE}}(\widetilde{\mathbf{X}}, y)\|^2}, & \text{otherwise} \end{cases} \tag{6}$$

Following PGD (Madry et al., 2017), we iteratively update $\mathbf{E}$ using an Adam optimizer (Kingma & Ba, 2014):

$$\mathbf{E}^{t+1} = \mathbf{E}^t + \eta \cdot Adam(\delta_{\mathbf{E}^t}) \tag{7}$$

where $\eta$ is the step size for each update.

### 3.6 SPARSE PATCH-FOOL: A SPARSE VARIANT OF PATCH-FOOL

**Motivation.** One natural question associated with Patch-Fool is: "How many pixels within a patch are needed to be perturbed for effectively misleading the model to misclassify the input image?". There exist two extreme cases: (1) perturbing only a few pixels that lead to local perturbations against which ViTs are more robust, and (2) perturbing the whole patch, i.e., our vanilla Patch-Fool. We hypothesize that answering this question helps better understand under what circumstances ViTs are more (or less) robust than CNNs. To this end, we study a variant of Patch-Fool, dubbed Sparse Patch-Fool, as defined below.

**Objective formulation.** For enabling Sparse Patch-Fool, we add a sparse constraint to Eq. 1, i.e.:

$$\underset{1 \le p \le n, \mathbf{E} \in \mathbb{R}^{n \times d}, \mathbf{M} \in \{0,1\}^{n \times d}}{\arg\max} J(\mathbf{X} + \mathbb{1}_p \odot (\mathbf{M} \circ \mathbf{E}), y) \ s.t. \ \|\mathbf{M}\|_0 \le k \tag{8}$$

where we use a binary mask $\mathbf{M}$ with a predefined sparsity parameter $k$ to control the sparsity of $\mathbf{E}$. To effectively learn the binary distribution of $\mathbf{M}$, we parameterize $\mathbf{M}$ as a continuous value $\widehat{\mathbf{M}}$, following (Ramanujan et al., 2020; Diffenderfer & Kailkhura, 2021). During forward, only the top $k$ highest elements of $\widehat{\mathbf{M}}$ is activated and set to 1 and others are set to 0 to satisfy the target sparsity constraint; and during backward, all the elements in $\widehat{\mathbf{M}}$ will be updated via straight-through estimation (Bengio et al., 2013). We jointly optimize $\widehat{\mathbf{M}}$ with $\mathbf{E}$ as in Eq. 7.

### 3.7 MILD PATCH-FOOL: A MILD VARIANT OF PATCH-FOOL

In addition to the number of perturbed pixels manipulated by Sparse Patch-Fool, the perturbation strength is another dimension for measuring the perturbations within a patch. We also propose a mild variant of Patch-Fool, dubbed Mild Patch-Fool, with a constraint on the norm of the perturbation $\mathbf{E}$ to ensure $\|\mathbf{E}\|_2 \le \epsilon$ or $\|\mathbf{E}\|_\infty \le \epsilon$ which are known as the $L_2$ and $L_\infty$ constraint, respectively. We achieve this by scaling (for the $L_2$ constraint) or clipping (for the $L_\infty$ constraint) $\mathbf{E}$ after updating it.

## 4 EVALUATION OF PATCH-FOOL

### 4.1 EVALUATION SETUP

**Models and datasets.** We mainly benchmark the robustness of the DeiT (Touvron et al., 2021) family with the ResNet (He et al., 2016) family, using their official pretrained models. Note that we adopt DeiT models without distillation for a fair comparison. We randomly select 2500 images from the validation set of ImageNet for evaluating robustness, following (Bhojanapalli et al., 2021).

**Patch-Fool settings.** The weight coefficient $\alpha$ in Eq. 4 is set as 0.002. The step size $\eta$ in Eq. 7 is initialized to be 0.2 and decayed by 0.95 every 10 iterations, and the number of total iterations is 250. For evaluating Patch-Fool with different perturbation strengths, we allow Patch-Fool to attack up to four patches based on the attention-aware patch selection in Sec. 3.4, i.e., the patches with top importance scores defined in Eq. 2 will be selected. Note that we report the robust accuracy instead of the attack success rate throughout this paper as our main focus is the robustness benchmark.

### 4.2 BENCHMARK THE ROBUSTNESS OF VITS AND CNNS AGAINST PATCH-FOOL

We adopt our Patch-Fool to attack ViTs and use the saliency map to guide the patch selection for attacking CNNs, which is the strongest attack setting as shown in Sec. 4.3. The resulting robust accuracy of both DeiT and ResNet families under different numbers of attacked patches is shown in Fig. 2. We can observe that DeiT models are consistently less robust against Patch-Fool than their ResNet counterparts under similar model complexity, e.g., compared with ResNet-50, DeiT-S suffers from a 16.31% robust accuracy drop under the single-patch attack of Patch-Fool, although it has a 3.38% and 18.70% higher clean and robustness accuracy against PGD-20 ($\epsilon = 0.001$), respectively. This indicates that ViTs are *not always* robust learners as they may underperform under customized perturbations as compared to CNNs and their seeming robustness against existing attacks can be overturned.

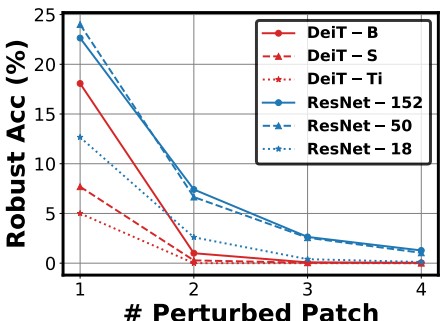

Figure 2: Benchmark the robustness of DeiTs and ResNets against Patch-Fool under different numbers of perturbed patches.

### 4.3 ABLATION STUDY: EFFECTIVENESS OF THE ATTENTION-AWARE PATCH SELECTION

To validate the effectiveness of our attention-aware patch selection method, we benchmark two variants of patch selection mechanism: (1) random patch selection, and (2) saliency-map-based patch selection. For the latter one, we adopt the averaged saliency score of a patch, defined as the averaged absolute value of the gradients on each pixel in a patch following (Simonyan et al., 2013), as the metric to select patches. For a fair comparison, we only adopt the final cross-entropy loss $J_{CE}$ in Eq. 4 in this set of experiments. As shown in Tab. 2, we can see that (1) among the three strategies for attacking ViTs, our attention-aware patch selection is the most effective strategy in most cases and thus we adopt it by default; (2) DeiT variants are still consistently less robust than their ResNet counterparts under similar model complexity, indi-

Table 1: Benchmark the robust accuracy of DeiTs and ResNets under different patch selection strategies, where 'xP' denotes a total of x patches are perturbed and the lowest robust accuracy is annotated in bold.

| Model | Selection Strategy | Robust Acc (%) | | | |
|---|---|---|---|---|---|
| | | **1P** | **2P** | **3P** | **4P** |
| DeiT-Ti | Random | 5.21 | 0.16 | **0.00** | **0.00** |
| | Saliency | 6.49 | **0.12** | **0.00** | **0.00** |
| | **Attn-score** | **4.01** | **0.12** | **0.00** | **0.00** |
| DeiT-S | Random | 7.41 | 0.36 | **0.00** | **0.00** |
| | Saliency | 11.70 | 0.32 | **0.00** | **0.00** |
| | **Attn-score** | **6.25** | **0.20** | **0.00** | **0.00** |
| DeiT-B | Random | 24.76 | 2.22 | 0.20 | **0.04** |
| | Saliency | 24.67 | 1.56 | 0.20 | **0.04** |
| | **Attn-score** | **22.12** | **1.32** | **0.16** | **0.04** |
| ResNet-18 | Random | 20.43 | 2.84 | 0.64 | **0.04** |
| | Saliency | **12.66** | **2.60** | **0.40** | 0.12 |
| ResNet-50 | Random | 31.57 | 9.66 | 3.12 | **0.60** |
| | Saliency | **24.00** | **6.65** | **2.56** | 1.04 |
| ResNet-152 | Random | 32.09 | 10.42 | **2.64** | **1.00** |
| | Saliency | **22.64** | **7.41** | **2.64** | 1.28 |

cating that attacking the basic component participating in self-attention calculations can indeed effectively degrade ViTs' robustness; and (3) Patch-Fool equipped with random patch selection, with a 2.64% robust accuracy gap against the best strategy, can already effectively degrade DeiTs'

robustness, while it cannot effectively attack ResNets without the guidance from the saliency map, indicating ViTs are generally more vulnerable than CNNs to patch-wise perturbations.

We also perform an ablation study for $l$ in Eq. 2 based on which layer the attention-aware patch selection is performed. As shown in Tab. 2, selecting the early layers generally achieves consistent better results than that of later layers, which we conjecture is because patches in early layers can still roughly maintain the original information extracted from the inputs while their

Table 2: Ablation study of the layer index for guiding the attention-aware patch selection. The resulting robust accuracy (%) is annotated in the table.

| $l$ | 1 | 2 | 3 | 4 | 5 | 6 |
|---|---|---|---|---|---|---|
| DeiT-B | 21.27 | 23.96 | 23.00 | 22.48 | 21.03 | 21.67 |

| $l$ | 7 | 8 | 9 | 10 | 11 | 12 |
|---|---|---|---|---|---|---|
| DeiT-B | 27.56 | 28.04 | 27.80 | 28.04 | 28.08 | 27.48 |

counterparts in later layers are mixed with information from other patches, leading to an inferior guidance for selecting the perturbed patch. This conjecture is validated by the observed phase change in the attention map, i.e., after the 6-th layer, more complexity correlations between patches are captured in addition to the diagonal ones. Therefore, we set $l = 5$ by default.

### 4.4 ABLATION STUDY: EFFECTIVENESS OF THE ATTENTION-AWARE LOSS

To evaluate the effectiveness of our attention-aware loss with the cosine-similarity-based re-weighting mechanism (see Sec. 3.5), we compare it with two baselines: (1) train with only the final cross-entropy loss, i.e., $J_{CE}$ is enabled without the attention-aware loss, and (2) $\beta_l = 0$, $\forall l \in [1, 12]$, i.e., the layerwise $J_{ATTN}^{(l)}$ in Eq. 4 is directly summed together with the final $J_{CE}$. As shown in Tab. 3, we can observe that (1) our attention-aware loss equipped with the cosine-

Table 3: Benchmark the attention-aware loss with two baselines, where 'Cos-sim' denotes our method of attention-aware loss with cosine-similarity-based re-weighting strategy while 'w/o' and 'Sum' are two baselines.

| Model | Atten-Loss | Robust Acc (%) | | | |
|---|---|---|---|---|---|
| | | 1P | 2P | 3P | 4P |
| DeiT-Base | w/o | 22.12 | 1.32 | 0.16 | **0.04** |
| | Sum | 63.46 | 16.11 | 1.96 | 0.10 |
| | **Cos-sim** | **18.95** | **1.00** | **0.08** | **0.04** |

similarity-based re-weighting strategy consistently achieves the best attack performance, e.g., a 3.17% reduction in robust accuracy compared with the baseline without attention-aware loss; and (2) directly summing up all the losses leads to poor convergence especially under limited perturbed patches.

### 4.5 BENCHMARK AGAINST SPARSE PATCH-FOOL

**Setup.** To study the influence of the sparsity of perturbed pixels for both CNNs and ViTs, we evaluate our proposed Sparse Patch-Fool via varying the global perturbation ratio (PR) of the whole image (i.e., $k$/total-pixel) as well as the number of patches allowed to be perturbed.

**Benchmark the robustness of ViTs and CNNs.** As shown in Tab. 4, under different perturbation ratios and numbers of perturbed patches, neither ViTs nor CNNs will always be the winner in robustness. In particular, under relatively small perturbation ratios or more perturbed patches (e.g., when all patches are allowed to be perturbed), CNNs will suffer from worse robustness, while ViTs will be more vulnerable learners under relatively large perturbation ratios as well as fewer perturbed patches.

**Influence of the number of perturbed patches.** We further study the influence of the number of perturbed patches under the same global perturbation ratio as shown in Tab. 5. We can see that (1) under a small perturbation ratio of 0.05%

Table 4: Benchmark the robust accuracy of DeiTs and ResNets against Sparse Patch-Fool under different perturbation ratios and perturbed patches, where 'All' denotes all patches are allowed to be perturbed and the lower robust accuracy is annotated in bold.

| PR | #Patch | DeiT-S | ResNet-50 | DeiT-B | ResNet-152 |
|---|---|---|---|---|---|
| 0.05% | 1P | 68.99 | **57.01** | 72.96 | **58.57** |
| | 2P | 67.15 | **51.8** | 71.35 | **54.61** |
| | 4P | 65.99 | **47.96** | 69.83 | **49.84** |
| | All | 68.39 | **49.2** | 67.03 | **53.93** |
| 0.10% | 1P | 59.74 | **48.96** | 65.26 | **48.12** |
| | 2P | 56.45 | **40.46** | 61.54 | **42.15** |
| | 4P | 52.20 | **34.05** | 58.89 | **36.38** |
| | All | 51.44 | **24.16** | 53.89 | **30.50** |
| 0.30% | 1P | 26.84 | **30.69** | 41.23 | **29.89** |
| | 2P | 21.47 | **20.99** | 32.17 | **21.51** |
| | 4P | 15.62 | **12.58** | 22.80 | **14.18** |
| | All | 14.02 | **0.76** | 16.55 | **2.24** |
| 0.40% | 1P | **16.63** | 26.52 | 32.13 | **25.12** |
| | 2P | **11.74** | 16.5 | 22.92 | **16.51** |
| | 4P | **7.57** | 9.46 | 12.14 | **10.10** |
| | All | 6.77 | **0.36** | 9.05 | **0.52** |
| 0.60% | 2P | **3.21** | 11.18 | **10.42** | 11.46 |
| | 4P | **1.68** | 5.57 | **4.09** | 6.01 |
| | All | 1.92 | **0.12** | 1.92 | **0.12** |
| 0.80% | 2P | **0.88** | 8.57 | **4.37** | 8.89 |
| | 4P | **0.32** | 3.69 | **1.00** | 4.25 |
| | All | 0.44 | **0.04** | 0.64 | **0.12** |

Table 5: Benchmark the robust accuracy of DeiTs and ResNets against Sparse Patch-Fool under a fixed perturbation ratio (PR=0.05%/0.5%) with different numbers of perturbed patches. The lower robust accuracy is annotated in bold.

| PR=0.05% | 1P | 2P | 4P | 8P | 16P | 32P | 64P | 128P | All |
|---|---|---|---|---|---|---|---|---|---|
| DeiT-S | 68.99 | 67.15 | 65.99 | 65.1 | 67.79 | 65.71 | 65.81 | 68.99 | 68.39 |
| ResNet-50 | **57.01** | **51.80** | **47.96** | **43.39** | **39.78** | **38.54** | **40.34** | **44.79** | **49.20** |
| DeiT-B | 72.96 | 71.35 | 69.83 | 69.67 | 68.95 | 68.71 | 67.91 | 68.51 | 67.03 |
| ResNet-152 | **58.57** | **54.61** | **49.84** | **46.03** | **43.99** | **43.83** | **44.67** | **49.40** | **53.93** |
| **PR=0.5%** | **1P** | **2P** | **4P** | **8P** | **16P** | **32P** | **64P** | **128P** | **All** |
| DeiT-S | **7.89** | **6.29** | **3.61** | **1.72** | 1.44 | 2.08 | 2.64 | 3.08 | 3.57 |
| ResNet-50 | 24.00 | 13.66 | 6.93 | 3.21 | **1.24** | **0.40** | **0.16** | **0.08** | **0.16** |
| DeiT-B | 24.24 | 17.11 | **7.33** | **2.40** | **1.68** | 1.88 | 2.88 | 3.73 | 4.81 |
| ResNet-152 | **21.75** | **14.22** | 8.09 | 3.93 | 1.88 | **0.96** | **0.24** | **0.16** | **0.20** |

Table 6: Benchmark the robust accuracy of DeiTs and ResNets against Patch-Fool under the $L_\infty$ constraint with different perturbation strengths $\epsilon$. Here vanilla Patch-Fool denotes the unconstrained Patch-Fool and the lower robust accuracy is annotated in bold.

| Model | Patch Num | Patch-Fool under $L_\infty$ Constraint | | | | | Vanila Patch-Fool |
|---|---|---|---|---|---|---|---|
| | | $\epsilon$=8/255 | $\epsilon$=16/255 | $\epsilon$=32/255 | $\epsilon$=64/255 | $\epsilon$=128/255 | |
| DeiT-Ti | 1P | 57.57 | 51.88 | 41.39 | 29.61 | **9.86** | **4.01** |
| | 2P | 54.65 | 34.62 | 14.7 | 4.45 | **0.56** | **0.12** |
| | 3P | 34.61 | 15.83 | **2.88** | **0.44** | **0.00** | **0.00** |
| | 4P | 25.08 | 6.65 | **0.56** | **0.00** | **0.00** | **0.00** |
| ResNet-18 | 1P | **45.31** | **39.66** | **28.89** | **19.79** | 13.10 | 12.66 |
| | 2P | **31.57** | **19.55** | **9.98** | **4.85** | 3.04 | 2.60 |
| | 3P | **19.71** | **8.97** | 3.57 | 1.08 | 0.68 | 0.40 |
| | 4P | **12.14** | **3.85** | 1.12 | 0.20 | 0.14 | 0.12 |
| DeiT-S | 1P | 66.83 | 60.62 | 45.83 | 30.77 | **14.50** | **6.25** |
| | 2P | 54.65 | 34.62 | 15.14 | 5.09 | **1.16** | **0.20** |
| | 3P | 40.31 | 17.43 | **3.45** | **0.52** | **0.04** | **0.00** |
| | 4P | 28.33 | 7.25 | **0.64** | **0.00** | **0.00** | **0.00** |
| ResNet-50 | 1P | **51.16** | **43.71** | **35.18** | **29.17** | 24.64 | 24.00 |
| | 2P | **36.70** | **23.16** | **14.90** | **10.38** | 7.65 | 6.65 |
| | 3P | **22.96** | **12.18** | 5.77 | 4.25 | 2.72 | 2.56 |
| | 4P | **15.02** | **5.85** | 2.44 | 1.12 | 0.41 | 1.04 |

which is closer to local perturbations, CNNs are the consistent loser in robustness; and (2) under a relatively large perturbation ratio of 0.5%, although increasing the number of perturbed patches leads to a consistent reduction in CNNs' robust accuracy, the robustness reduction for ViTs will quickly saturate, i.e., ViTs gradually switch from the loser to the winner in robustness as compared to CNNs.

**Insights.** We analyze that smaller perturbation ratios under the same number of perturbed patches or more perturbed patches under the same perturbation ratio will lead to less perturbed pixels within one patch, i.e., a lower *perturbation density*, which is closer to local perturbations against which ViTs are thus more robust than CNNs. In contrast, given more perturbed pixels in one patch, i.e., a higher perturbation density for which an extreme case is our vanilla Patch-Fool, ViTs become more vulnerable learners than CNNs. This indicates that a high perturbation density can be a perspective for exploring ViTs' vulnerability, which has been neglected by existing adversarial attacks.

Considering the perturbation strength is another dimension for measuring the perturbations within a patch in addition to the perturbation density, we evaluate our proposed Mild Patch-Fool in Sec. 4.6.

## 4.6 BENCHMARK AGAINST MILD PATCH-FOOL

**Setup.** To study the influence of the perturbation strength within each patch, we evaluate our proposed Mild Patch-Fool in Sec. 4.6 with $L_2$ or $L_\infty$ constraints on the patch-wise perturbations with

different strengths indicated by $\epsilon$. Note that the perturbation strength $\epsilon$ of $L_2$-based Mild Patch-Fool is summarized over all perturbed patches. We benchmark both the DeiT and ResNet families with different numbers of perturbed patches as shown in Tab. 6 and Tab. 7.

**Observations and analysis.** We can observe that (1) the robust accuracy will be degraded more by larger perturbation strength indicated by $\epsilon$ under both $L_2$ and $L_\infty$ constraints, and (2) more importantly, DeiTs are more robust than ResNets under small $\epsilon$, and gradually become more vulnerable than ResNets as $\epsilon$ increases. For example, as gradually increasing $\epsilon$ from $8/255$ to $128/255$ under $L_\infty$ attacks, DeiT-S switches from the winner to the loser in robustness as compared to ResNet-50.

**Insights.** This set of experiments, together with the analysis in Sec. 4.5, reflects that the perturbation density and the perturbation strength are two key determinators for the robustness ranking between ViTs and CNNs: higher/lower perturbation density or perturbation strength will make ViTs the loser/winner in robustness. This first-time finding can enhance the understanding about the robustness ranking between ViTs and CNNs, and aid the decision making about which models to deploy in different real-world scenarios with high security awareness.

Table 7: Benchmark the robust accuracy of DeiTs and ResNets against Patch-Fool under the $L_2$ constraint with different perturbation strengths $\epsilon$, which is summarized over all perturbed patches. The lower robust accuracy is annotated in bold.

| Model | Patch Num | Patch-Fool under $L_2$ Constraint | | | | | Vanila Patch-Fool |
|---|---|---|---|---|---|---|---|
| | | $\epsilon$=0.5 | $\epsilon$=1 | $\epsilon$=2 | $\epsilon$=4 | $\epsilon$=6 | |
| DeiT-Ti | 1P | 58.05 | 48.36 | 38.5 | **17.75** | **5.61** | **4.01** |
| | 2P | 51.84 | 34.62 | 16.31 | **4.45** | **0.80** | **0.12** |
| | 3P | 47.24 | 25.40 | 7.85 | **1.04** | **0.08** | **0.00** |
| | 4P | 44.27 | 19.39 | 4.09 | **0.02** | **0.00** | **0.00** |
| ResNet-18 | 1P | **44.91** | **32.93** | **23.40** | 18.22 | 13.1 | 12.66 |
| | 2P | **33.81** | **19.55** | **9.05** | 4.57 | 2.92 | 2.60 |
| | 3P | **27.56** | **11.14** | **3.85** | 1.32 | 0.44 | 0.40 |
| | 4P | **22.04** | **6.69** | **1.88** | 0.36 | 0.24 | 0.12 |
| DeiT-S | 1P | 67.23 | 56.29 | 41.23 | **22.84** | **9.74** | **6.25** |
| | 2P | 60.98 | 40.62 | 17.59 | **4.57** | **1.20** | **0.20** |
| | 3P | 56.53 | 30.57 | 7.73 | **1.24** | **0.20** | **0.00** |
| | 4P | 53.53 | 24.60 | 3.77 | **0.28** | **0.04** | **0.00** |
| ResNet-50 | 1P | **49.84** | **37.74** | **29.21** | 24.16 | 24.09 | 24.00 |
| | 2P | **35.66** | **20.63** | **12.06** | 7.93 | 7.25 | 6.65 |
| | 3P | **28.04** | **12.54** | **6.13** | 2.92 | 2.60 | 2.56 |
| | 4P | **22.76** | **8.13** | **3.04** | 1.28 | 1.16 | 1.04 |

We also benchmark the effectiveness of Patch-Fool on top of adversarial trained ViTs/CNNs, evaluate the patch-wise adversarial transferability of Patch-Fool, and visualize the adversarial examples generated by our Patch-Fool's different variants in Appendix. A.2~ A.4, respectively.

## 5 CONCLUSION

The recent breakthroughs achieved by ViTs in various vision tasks have attracted an increasing attention on ViTs' robustness, aiming to fulfill the goal of deploying ViTs into real-world vision applications. In this work, we provide a new perspective regarding ViTs' robustness and propose a novel attack framework, dubbed Patch-Fool, to attack the basic component (i.e., a single patch) in ViTs' self-attention calculations, against which ViTs are found to be more vulnerable than CNNs. Interestingly, the proposed Sparse Patch-Fool and Mild Patch-Fool attacks, two variants of our Patch-Fool, further indicate that the perturbation density and perturbation strength onto each patch seem to be the two key factors that determine the robustness ranking between ViTs and CNNs. We believe this work can shed light on better understanding ViTs' robustness and inspire innovative defense techniques.

## ACKNOWLEDGEMENT

The work is supported by the National Science Foundation (NSF) through the MLWiNS program (Award number: 2003137), an NSF CAREER award (Award number: 2048183), and the RTML program (Award number: 1937592).

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

# A APPENDIX

## A.1 EVALUATING ROBUSTNESS OF VITS AND CNNS UNDER EXISTING ATTACKS

Although various comparisons on the robustness of ViTs and CNNs have been explored in pioneering works (Bhojanapalli et al., 2021; Aldahdooh et al., 2021; Shao et al., 2021), their evaluation suffers from one of the following limitations: (1) only adopt weak attack methods, (2) only adopt early ViT designs without considering recently advanced ViT architectures, and (3) do not adopt the official and latest pretrained models and suffer from inferior clean accuracies. To this end, we extensively evaluate the robustness against common white-box attacks of several representative ViT variants, which cover the popular trends in designing ViT architectures, including (1) using local self-attention (Swin (Liu et al., 2021a)), which adopts the attention mechanism within a local region instead of the global ones in vanilla ViTs to capture low-level features and reduce the computational cost, and (2) introducing the inductive bias of CNNs to build hybrid models (LeViT (Graham et al., 2021)).

Table 8: Benchmark the robustness of three ViT families and two CNN families against PGD attacks (Madry et al., 2017), Auto-Attack (Croce & Hein, 2020), and CW attacks (Carlini & Wagner, 2017), where the perturbation strengths of the $L_\infty$ attacks are annotated.

| Model | FLOPs/ Params | Clean Acc (%) | PGD-20 (%) | | | Auto-Attack (%) | | CW-$L_2$ (%) | CW-$L_\infty$ (%) | |
| --- | --- | --- | --- | --- | --- | --- | --- | --- | --- | --- |
| | | | 0.001 | 0.003 | 0.005 | 0.001 | 0.003 | | 0.003 | 0.005 |
| ResNet-18 | 1.80G/23M | 71.78 | 22.60 | 0.34 | 0.00 | 22.29 | 0.06 | 9.95 | 9.10 | 5.31 |
| ResNet-50 | 4.11G/98M | 76.15 | 35.00 | 1.42 | 0.18 | 25.42 | 1.23 | 10.80 | 13.39 | 7.95 |
| ResNet-101 | 7.83G/170M | 78.25 | 37.20 | 2.10 | 0.42 | 27.92 | 1.88 | 11.19 | 15.36 | 11.11 |
| ResNet-152 | 11.56G/230M | 78.31 | 42.09 | 2.56 | 0.24 | 39.58 | 2.34 | 15.20 | 17.24 | 14.58 |
| VGG-16 | 15.3G/134M | 71.30 | 23.24 | 0.28 | 0.02 | 15.42 | 0.04 | 8.15 | 7.84 | 4.84 |
| DeiT-Ti | 1.26G/5M | 72.02 | 41.51 | 6.73 | 1.52 | 38.29 | 5.75 | 30.08 | 28.74 | 21.05 |
| DeiT-S | 4.61G/22M | 79.53 | 53.70 | 11.40 | 2.91 | 51.88 | 7.70 | 43.78 | 44.22 | 34.65 |
| DeiT-B | 17.58G/86M | 81.90 | 52.20 | 10.10 | 2.28 | 51.08 | 8.62 | 43.77 | 47.40 | 36.35 |
| Swin-T | 4.51G/29M | 80.96 | 38.80 | 2.72 | 1.08 | 35.81 | 2.04 | 28.84 | 35.10 | 20.63 |
| Swin-S | 8.77G/50M | 82.37 | 47.20 | 6.79 | 0.42 | 43.25 | 4.58 | 30.32 | 41.22 | 28.91 |
| Swin-B | 15.47G/88M | 84.48 | 47.16 | 7.30 | 1.14 | 44.58 | 6.52 | 34.34 | 43.57 | 34.00 |
| LeViT-256 | 1.13G/19M | 81.60 | 36.42 | 3.00 | 0.64 | 31.25 | 2.29 | 43.94 | 46.57 | 25.58 |
| LeViT-384 | 2.35G/39M | 82.60 | 42.19 | 3.08 | 0.24 | 35.83 | 2.50 | 47.15 | 50.22 | 39.56 |

### A.1.1 EVALUATION SETUP

**Models and datasets.** We evaluate the robustness of three ViT families (i.e., DeiT (Touvron et al., 2021), Swin (Liu et al., 2021a), and LeViT (Graham et al., 2021)) and two CNN families (ResNet (He et al., 2016) and VGG (Simonyan & Zisserman, 2014)) on ImageNet using their official implementation and pretrained models. Note that we adopt DeiT models without distillation, which only improves the training schedule over vanilla ViTs, for a fair comparison.

**Attack settings.** We adopt four adversarial attacks (i.e., PGD (Madry et al., 2017), AutoAttack (Croce & Hein, 2020), CW-$L_\infty$ (Carlini & Wagner, 2017), and CW-$L_2$) with different perturbation strengths. In particular, for the CW-$L_\infty$ and CW-$L_2$ attacks, we adopt the implementation in AdverTorch (Ding et al., 2019) and the same settings as (Chen et al., 2021a; Rony et al., 2019); For AutoAttack, we adopt the official implementation and default settings in (Croce & Hein, 2020).

### A.1.2 OBSERVATIONS AND ANALYSIS

**Observations.** From the evaluation results summarized in Tab. 8, we make the following observations: (1) ViTs are consistently more robust than CNNs with comparable model complexities under all attack methods, which is consistent with the previous observations (Bhojanapalli et al., 2021; Aldahdooh et al., 2021; Shao et al., 2021). In particular, DeiT-S/DeiT-B achieves a 18.70%/10.11% higher robust accuracy over ResNet-50/ResNet-152 under PGD-20 attacks with a perturbation strength of 0.001; (2) compared with vanilla ViTs, ViT variants equipped with local self-attention or convolutional modules, which improves the model capability to capture local features and thus boosts the clean accuracy, are more vulnerable to adversarial attacks, although they are still more robust than CNNs with comparable complexities. For example, Swin-T/Swin-B suffers from a 14.90%/5.04% robust accuracy drop compared with DeiT-S/DeiT-B under PGD-20 attacks with a perturbation strength of 0.001; and (3) the degree of overparameterization has less influence in the robustness for the same

family of ViT models compared with its great influence in CNNs' robustness, as the most lightweight DeiT-Ti can already achieve a comparable robust accuracy (-0.58%) as ResNet-152, while requiring $9.17\times/46\times$ less floating-point operations (FLOPs)/parameters.

**Analysis.** Combining the three insights drawn from the aforementioned observations, we can observe the superiority of the global attention mechanism over convolutional and local self-attention blocks, in terms of both improved robust accuracy and reduced sensitivity to the degree of model overparameterization. This indicates that the global attention mechanism itself can serve as a good robustification technique against existing adversarial attacks, even in lightweight ViTs with small model complexities. For example, as shown in Fig. 1, the gap between the attention maps generated by clean and adversarial inputs in deeper layers remains small. We wonder that "Are the global attentions in ViTs truly robust, or their vulnerability has not been fully explored and exploited?". To answer this, we propose our customized attack Patch-Fool in Sec. 3 and find that the vulnerability of global attentions can be utilized to degrade the robustness of ViTs, making them more vulnerable learners than CNNs.

### A.2 PATCH-FOOL ON TOP OF ADVERSARIALLY TRAINED MODELS

To study the influence of robust training algorithms against our Patch-Fool, we further benchmark the robustness of both adversarially trained ViTs and CNNs.

**Setup.** We apply Fast Adversarial Training (FAT) (Wong et al., 2019) with an $\epsilon$ of 2/255 and 4/255 under the $L_\infty$ constraint on top of both DeiT-Ti and ResNet-18 on ImageNet. We report the robust accuracy of the FAT trained models against our Patch-Fool in Tabs. 9 and 10.

**Observations and analysis.** From Tab. 9, we can observe that although FAT improves the robustness of both DeiT-Ti and ResNet-18 against our Patch-Fool attacks, DeiT-Ti is still more vulnerable against Patch-Fool than ResNet-18 under the same number of perturbed patches. In addition, we can observe from Tab. 10 that (1) stronger adversarial training with larger $\epsilon$ leads to better robustness against both PGD attacks and our Patch-Fool, and (2) the improvement in robust accuracy against PGD attacks is higher than the one against Patch-Fool, indicating that enhanced adversarial training schemes or other defense methods are required to robustify ViTs against our Patch-Fool, which is also our future work.

Table 9: Benchmark the robustness of adversarially trained DeiT-Ti and ResNet-18 against both PGD attacks (Madry et al., 2017) and Patch-Fool with different numbers of perturbed patches. Here "w/o" denotes the results without any adversarial training.

| Model | FAT | Clean Acc (%) | PGD $\epsilon$=0.2/255 | PGD $\epsilon$=2/255 | Patch-Fool | | | |
|---|---|---|---|---|---|---|---|---|
| | | | | | 1P | 2P | 3P | 4P |
| DeiT-Ti | w/o | 72.02 | 44.11 | 0.08 | 4.01 | 0.12 | 0 | 0 |
| | $\epsilon$=4/255 | 66.35 | 64.46 | 41.83 | **22.48** | **6.25** | **2.64** | **0.72** |
| ResNet-18 | w/o | 71.78 | 30.49 | 0 | 12.66 | 2.60 | 0.10 | 0.12 |
| | $\epsilon$=4/255 | 60.13 | 58.57 | 37.18 | 23.48 | 11.58 | 6.09 | 2.69 |

Table 10: Evaluating the robustness of DeiT-Ti adversarially trained by different perturbation strengths against both PGD attacks (Madry et al., 2017) and Patch-Fool with different numbers of perturbed patches. Here "w/o" denotes the results without any adversarial training.

| Model | FAT | Clean Acc (%) | PGD $\epsilon$=0.2/255 | PGD $\epsilon$=2/255 | Patch-Fool | | | |
|---|---|---|---|---|---|---|---|---|
| | | | | | 1P | 2P | 3P | 4P |
| DeiT-Ti | w/o | 72.02 | 44.11 | 0.08 | 4.01 | 0.12 | 0 | 0 |
| | $\epsilon$=2/255 | 67.43 | 63.26 | 40.06 | 19.75 | 5.85 | 1.88 | 0.52 |
| | $\epsilon$=4/255 | 66.35 | 64.46 | 41.83 | 22.48 | 6.25 | 2.64 | 0.72. |

### A.3 PATCH-WISE ADVERSARIAL TRANSFERABILITY OF PATCH-FOOL

We further discuss the patch-wise adversarial transferability of Patch-Fool, i.e., transfer the perturbations generated for attacking one specific patch to attack other patches on the same image.

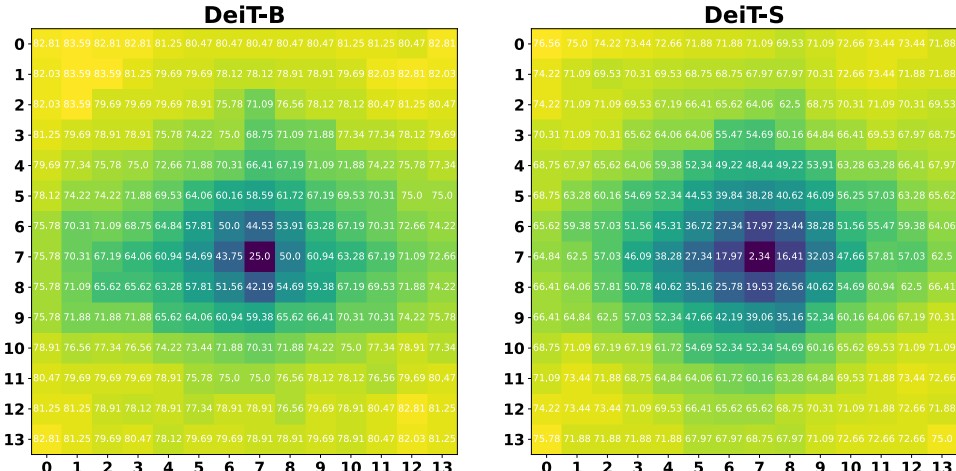

Figure 3: Visualizing the patch-wise adversarial transferability of Patch-Fool on top of DeiT-B (left) and DeiT-S (right), where the robust accuracy when perturbing each patch with the attack generated for the center patch on the same image is annotated in the figure.

**Setup.** Without losing generality, we generate the adversarial perturbation for the center patch with Patch-Fool which is adopted to attack all other patches on the same image and the resulting robust accuracy is annotated in Fig. 3. We average the robust accuracy at each patch location over a batch of 128 images.

**Observations.** We can observe that the adversarial patches generated by Patch-Fool can be transferred to neighboring patches with more notable accuracy degradation, while the adversarial transferability between patches far away from each other is poor.

### A.4 VISUALIZING THE ADVERSARIAL EXAMPLES GENERATED BY PATCH-FOOL'S VARIANTS

Here we visualize the adversarial examples generated by Patch-Fool's variants in Fig. 4, including (1) Patch-Fool with different number of perturbed patches (rows 2∼3), (2) Sparse Patch-Fool with a total of 250 perturbed pixels distributed in different number of perturbed patches (rows 4∼6), and (3) Mild Patch-Fool under $L_2$ and $L_\infty$ constraints (rows 7∼8). The corresponding robust accuracy is also annotated.

**Observations.** From the aforementioned visualization in Fig. 4, we can observe that (1) the adversarial patches generated by Patch-Fool visually resemble and emulate natural corruptions in a small region of the original image caused by potential defects of the sensors or potential noises/damages of the optical devices (see row 2), (2) more perturbed patches lead to a lower robust accuracy and worse imperceptibility (see row 3), (3) the generated adversarial perturbations of our Sparse Patch-Fool resemble impulse noises, which improves imperceptibility while still notably degrading the robust accuracy especially when perturbing more patches (see rows 4∼6), and (4) adding $L_2$ and $L_\infty$ constraints will notably improve the imperceptibility while incurring less degradation in the robust accuracy (rows 7∼8).

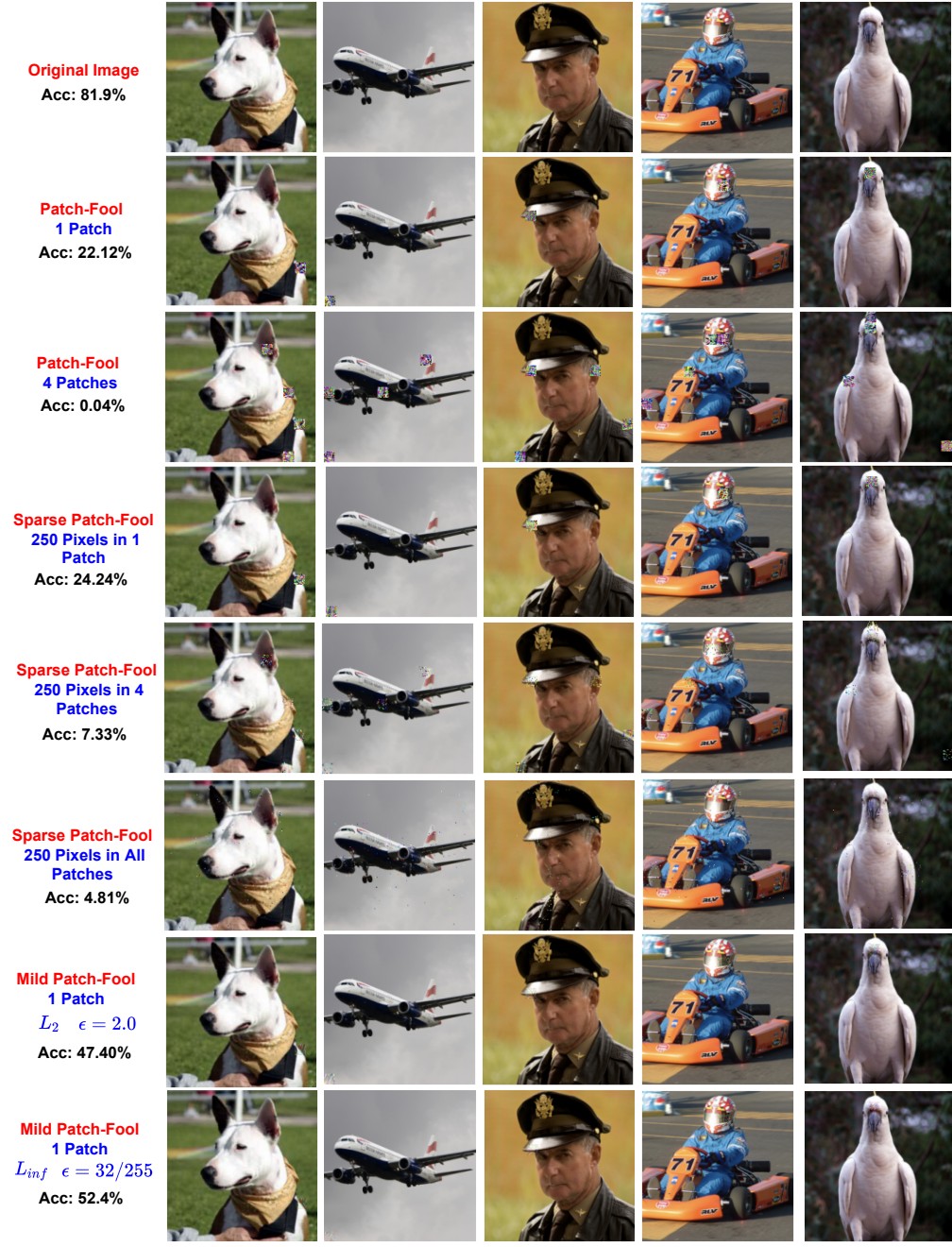

Figure 4: Visualizing the adversarial examples generated by Patch-Fool's variants, including Patch-Fool with different number of perturbed patches (rows 2∼3), Sparse Patch-Fool with a total of 250 perturbed pixels distributed in different number of perturbed patches (rows 4∼6), and Patch-Fool under $L_2$ and $L_\infty$ constraint (rows 7∼8). Note that both the attack settings and the resulting robust accuracy are annotated.

