# OpenReview forum: "Patch-Fool: Are Vision Transformers Always Robust Against Adversarial Perturbations?"
_ICLR.cc/2022/Conference — ICLR 2022 Poster_

### Official Review · Reviewer_r33g · 2021-10-30

**Correctness:** 4
**Technical Novelty And Significance:** 4
**Empirical Novelty And Significance:** 4
**Recommendation:** 8
**Confidence:** 4

**Main Review:**

Strengths:
1. This paper provides a new perspective for evaluating the robustness of ViTs which is novel and insightful.
2. From my point of view, the conclusion that the perturbation density is the key factor that influences the robustness ranking between ViTs and CNNs is significant to this field and can provide a better understanding of ViTs’ robustness.
3. This paper is well-written and easy to follow. The experiments are extensive and solid enough.

Weaknesses:
1. I notice that the robustness benchmarks of this paper are under the non-robust models. Some recent works have also proposed adversarial training for ViTs such as [1]. Can the authors provide the results on adversarial trained ViTs and CNNs for better benchmarking the robustness of these models?

[1] Shao, Rulin, et al. "On the adversarial robustness of visual transformers." arXiv preprint arXiv:2103.15670 (2021).


**Summary Of The Paper:**

This paper proposes a Patch-Fool attack which fools the self-attention mechanism of ViTs to evaluate the robustness of ViTs and CNNs-based models. Prior works investigated the robustness of ViTs and CNNs under the adversarial attacks designed mainly for CNNs and conjectured that ViTs are more robust than CNNs. However, in this paper, the authors attack a specific single patch and develop an attention-aware attack framework against which they find that ViTs are weaker learners than CNNs. Moreover, by developing Spare Patch-Fool, the authors find that ViTs are more vulnerable than CNNs under high perturbation density.


**Summary Of The Review:**

This paper proposes an effective Patch-Fool Attack for evaluating the robustness of ViTs, which is novel and the key finding of this paper is significant. I suggest acceptance.

---

> ### Author Response · Authors · 2021-11-23
> **Response to Reviewer r33g**
>
> Thanks for recognizing the  significance of our work for understanding the robustness ranking between ViTs and CNNs. We have addressed your comments as follows and provided more experiments and analysis in the appendix of our updated manuscript.
>
> Following your valuable suggestion, we further benchmark the robustness of adversarially trained ViTs and CNNs, and still we consistently observe the inferior robustness of ViTs against our proposed Patch-Fool attack. In particular, we apply Fast Adversarial Training (FAT) [1] with an $\epsilon$ of 2/255 and 4/255 under $L_{inf}$ constraint on top of DeiT-Ti and ResNet-18 on ImageNet. We report the robust accuracy of the FAT trained models against our Patch-Fool (“xP’' denotes the number of perturbed patches and "w/o" denotes the results without any adversarial training as reported in our submitted manuscript) in the table below. We can observe that although FAT improves the robustness of both DeiT-Ti and ResNet-18 against our Patch-Fool attacks, DeiT-Ti is still more vulnerable against Patch-Fool than ResNet-18 under the same number of perturbed patches.
>
>
> |  |  |  | PGD-$L_{inf}$ | PGD-$L_{inf}$ | Patch-Fool | Patch-Fool | Patch-Fool | Patch-Fool |
> |:---:|:---:|:---:|:---:|:---:|:---:|:---:|:---:|:---:|
> | **Model** | **FAT** | **Clean Acc (%)** |  **$\epsilon$=0.2/255** |  **$\epsilon$=2/255** | **1P** | **2P** | **3P** | **4P** |
> | DeiT-Ti | w/o | 72.02 | 44.11 | 0.08 | 4.01 | 0.12 | 0 | 0 |
> |  | $\epsilon$=4/255 | 66.35 | 64.46 | 41.83 | 22.48 | 6.25 | 2.64 | 0.72 |
> | ResNet-18 | w/o | 71.78 | 30.49 | 0 | 12.66 | 2.60 | 0.10 | 0.12 |
> |  | $\epsilon$=4/255 | 60.13 | 58.57 | 37.18 | 23.48 | 11.58 | 6.09 | 2.69 |
>
>
> In addition, we further do more ablation studies on FAT trained DeiT-Ti against PGD attacks and our Patch-Fool as shown in the table below. We can observe that (1) stronger adversarial training with larger $\epsilon$ leads to better robustness against both PGD attacks and our Patch-Fool, and (2) the improvement in robust accuracy against PGD attacks is higher than that against Patch-Fool, indicating that enhanced adversarial training schemes or other defense methods are required to robustify ViTs against our Patch-Fool, which is also our future work.
>
> |  |  |  | PGD-$L_{inf}$ | PGD-$L_{inf}$ | Patch-Fool | Patch-Fool | Patch-Fool | Patch-Fool |
> |:---:|:---:|:---:|:---:|:---:|:---:|:---:|:---:|:---:|
> | **Model** | **FAT** | **Clean Acc (%)** |  **$\epsilon$=0.2/255** | **$\epsilon$=2/255** | **1P** | **2P** | **3P** | **4P** |
> | DeiT-Ti | w/o | 72.02 | 44.11 | 0.08 | 4.01 | 0.12 | 0 | 0 |
> |  | $\epsilon$=2/255 | 67.43 | 63.26 | 40.06 | 19.75 | 5.85 | 1.88 | 0.52 |
> |  | $\epsilon$=4/255 | 66.35 | 64.46 | 41.83 | 22.48 | 6.25 | 2.64 | 0.72 |
>
> We will include such discussions in the final version and add more benchmarks on top of more adversarially trained models given more time.
>
> [1] “Fast is better than free: Revisiting adversarial training”, E. Wong et al., ICLR 2020.

---

> > ### Comment · Reviewer_r33g · 2021-11-27
> > **Thank you for the response**
> >
> > Thank you for the response and the additional experimental results. I think the robustness benchmark on adversarially trained models against Patch-Fool can give a more comprehensive evaluation. Besides, the results reveal that more adversarial training methods for ViTs can be developed.

---

### Official Review · Reviewer_L4v1 · 2021-11-02

**Correctness:** 3
**Technical Novelty And Significance:** 3
**Empirical Novelty And Significance:** 2
**Recommendation:** 6
**Confidence:** 4

**Main Review:**

To my knowledge, the approach is novel and performance gains (degradation in this case) non-trivial. Overall, the paper is fairly easy to understand and the evaluation is fair.

Here are some of my thoughts/criticisms:
1. My main concern is wrt to the lack of example of images of adversarial examples. From whaat I have seen, there is only one image in Fig 1 which shows the output of the technique proposed. The authors later go on to show that increasing the number of patches seems to make the attack more effective. However, at the point, is the image even adversarial in the traditional definition of the word? (i.e.  change in image is imperceptible to human eyes). Patch-Fool does not have an \epsilon factor like traditional robust training seems to have, so, essentially, we can perturb the image by a large amount to make the attack more potent and potentially make large changes to the image. Maybe the authors could also quantify how far these adversarial examples are via L2 distance.
2. Related to the previous point, if we perturb all patches in the image, accuracy drops drastically for both CNNs and ViTs. So, can this be considered a procedure to generate adversarial examples for both architectures simultaneously? The authors partially answer this in the sparse case in Table 7 but I was curious about the full perturbation case. Again, without an example image to look at, it is hard to say whether all patches being changed is even an adversarial example in aforementioned sense.
3. What are the numbers in table 3? I realized its accuracy on a later reading but the authors should mention this in the caption
4. A minor point about terminology - I have seen “Robust accuracy” used in the literature to mean accuracy of a robust model (i.e. model trained via robust training) whereas here, I assume, the authors used a model trained on clean data and fed it adversarial images. This seems to be implied in the paper but would be useful to mention to avoid confusion.

Formatting/Typos:
1. Section 4.3 - “according a predefined value l” should be  “according to a predefined value l”
2. The title for section 5.2 is the last line of the page. I understand the authors are trying to adhere to page limits but it breaks the “flow” readability-wise. I would encourage the authors to correct this.

**Summary Of The Paper:**

This paper propose a new attack on Vision Transformers (ViTs) called Patch-Fool. The attack proceeds by first picking a patch which contributes the most (in the self attention calculation) to other patches and then perturbed it adversarially wrt cross entropy + attention based loss. The results show that this kind of attack degrades performance significantly wrt prior work. The authors then perform various ablation studies to justify their architecture/loss choices.

**Summary Of The Review:**

Ultimately, I think there is a good idea here but the lack of qualitative examples makes a fair evaluation of this technique not possible. I would not recommend publication of this manuscript in its current state.

---

> ### Author Response · Authors · 2021-11-23
> **Response to Reviewer L4v1: Part 1**
>
> Thanks for recognizing the novelty and performances of our work. We have addressed your concerns as follows and provided more experiments and analysis in the appendix of our updated manuscript.
>
> **1. Visualization of the generated adversarial examples**
>
> Thanks for your suggestion! We have added the visualization of the generated adversarial examples by different variants of Patch-Fool in the updated manuscript (see Figure 3 in Appendix A.2 on the page 17 of our updated manuscript).
>
> First of all, we humbly clarify that as recognized by Reviewers gMrG and r33g, the main focus of our work is to provide a novel perspective about the robustness ranking between ViTs and CNNs, thus we mainly benchmark their quantitative performances, i.e., robust accuracy against patch-wise adversarial perturbations.
>
> In addition, we have added more visualizations of the adversarial examples in Figure 3 / Appendix A.2, for which the detailed settings can be found in the caption. From those examples, we can observe that (1) the adversarial patches generated by Patch-Fool visually resemble and emulate natural corruptions in a small region of the original image caused by potential defects of the sensors or potential noises/damages of the optical devices (row 2), (2) more perturbed patches lead to a lower robust accuracy and worse imperceptibility (row 3), and (3) the generated adversarial perturbations of our Sparse Patch-Fool resemble impulse noises, which improves imperceptibility while still notably degrading the robust accuracy especially when perturbing more patches (rows 4~6).
>
> As you suggested and to further enhance the imperceptibility of our Patch-Fool, we further add $L_2$ and $L_{inf}$ constraints to the perturbations generated by Patch-Fool. We add the corresponding visualization in rows 7~8 of Figure 3 and report the robust accuracy of DeiT-S/ResNet-50 in the table below. Under both $L_2$ and $L_{inf}$ constraints, we can observe that (1) the robust accuracy will be degraded more by larger perturbation strength indicated by $\epsilon$, and (2) more importantly, DeiT-S is more robust than ResNet-50 under small $\epsilon$, and gradually becomes more vulnerable than ResNet-50 as $\epsilon$ increases, which reflects another perspective of the perturbation density. In particular, in addition to the number of pixels perturbed in one patch analyzed in our manuscript, perturbation strength of each pixel is another perspective of perturbation density, and we consistently observe ViTs are more vulnerable than CNNs under larger perturbation density in terms of both the number of perturbed pixels and perturbation strength per pixel. More related results and analysis can be found in Appendix A.1 and we will add more discussions to the final version as we believe this will further strengthen our work.
>
> |  |  | $L_2$-based  Patch-Fool | $L_2$-based  Patch-Fool | $L_2$-based  Patch-Fool | $L_2$-based  Patch-Fool |  |
> |:---:|:---:|:---:|:---:|:---:|:---:|:---:|
> | **Model** | **Patch** | **$\epsilon$=0.5** | **$\epsilon$=1** | **$\epsilon$=2** | **$\epsilon$=4** | **Vanila Patch-Fool** |
> | DeiT-S | 1P | 67.23 | 56.29 | 41.23 | **22.84** | **6.25** |
> |  | 2P | 60.98 | 40.62 | 17.59 | **4.57** | **0.20** |
> |  | 3P | 56.53 | 30.57 | 7.73 | **1.24** | **0** |
> |  | 4P | 53.53 | 24.60 | 3.77 | **0.28** | **0** |
> | ResNet-50 | 1P | **49.84** | **37.74** | **29.21** | 24.16 | 24.00 |
> |  | 2P | **35.66** | **20.63** | **12.06** | 7.93 | 6.65 |
> |  | 3P | **28.04** | **12.54** | **6.13** | 2.92 | 2.56 |
> |  | 4P | **22.76** | **8.13** | **3.04** | 1.28 | 1.04 |
>
>
> |  |  |  |  | $L_{inf}$-based  Patch-Fool | |  |  |
> |:---:|:---:|:---:|:---:|:---:|:---:|:---:|:---:|
> | **Model** | **Patch** | **$\epsilon$=8/255** | **$\epsilon$=16/255** | **$\epsilon$=32/255** | **$\epsilon$=64/255** | **$\epsilon$=128/255** | **Vanila Patch-Fool** |
> | DeiT-S | 1P | 66.83 | 60.62 | 45.83 | 30.77 | **14.5** | **6.25** |
> |  | 2P | 54.65 | 34.62 | 15.14 | **5.09** | **1.16** | **0.20** |
> |  | 3P | 40.31 | 17.43 | **3.45** | **0.52** | **0.04** | **0** |
> |  | 4P | 28.33 | 7.25 | **0.64** | **0** | **0** | **0** |
> | ResNet-50 | 1P | **51.16** | **43.71** | **35.18** | **29.17** | 24.64 | 24.00 |
> |  | 2P | **36.70** | **23.16** | **14.9** | 10.38 | 7.65 | 6.65 |
> |  | 3P | **22.96** | **12.18** | 5.77 | 4.25 | 2.72 | 2.56 |
> |  | 4P | **15.02** | **5.85** | 2.44 | 1.12 | 0.41 | 1.04 |

---

> > ### Author Response · Authors · 2021-11-23
> > **Response to Reviewer L4v1: Part 2**
> >
> > **2. Perturb all the patches with Sparse Patch-Fool**
> >
> > Yes, you are right! Our Sparse Patch-Fool can be considered as a general procedure to generate adversarial examples when perturbing all patches, which belongs to one kind of sparse attack.  In Table 7 of our submitted manuscript, we perturb all patches with different perturbation ratios (i.e., ratio of the perturbed pixels) and benchmark with an SOTA sparse attack PGD0 and find that our Sparse Patch-Fool can achieve better attacking performances. We have added the visualization of perturbing all patches (totally 250 pixels) with Sparse Patch-Fool in row 6 of Figure 3 (Appendix A.2).
> >
> > **3. Numbers in Tab. 3**
> >
> > Yes, you are right, It is the robust accuracy. Thanks for pointing it out and we have added this information in the updated manuscript.
> >
> > **4. The terminology “robust accuracy”**
> >
> > We follow recent works [1][2] to use “robust accuracy” to denote the accuracy evaluated under adversarial examples even if the model is not adversarially trained. We will follow your suggestion to clarify this in the final version.
> >
> > [1] “On The Adversarial Robustness of Vision Transformers”, R. Shao et al., arXiv 2021.
> > [2] “Towards Robust Vision Transformer”, X. Mao et al., arXiv 2021.
> >
> > **5. Typo and formatting issues**
> >
> > Thanks for pointing these out! We have revised the typos and formats in the updated manuscript and will further proof-read in the final version.

---

> > ### Comment · Reviewer_L4v1 · 2021-11-27
> > **Thanks**
> >
> > Thank you for the additional experiments and visualizations, I think they illustrate the point being made in the paper further. Also, as you mentioned, I probably slightly misunderstood the main intent of the paper (primarily studying robustness vs a new attack technique). Thanks for the clarification.
> >
> > I have updated my score accordingly (to a 6).

---

### Official Review · Reviewer_gMrG · 2021-11-02

**Correctness:** 4
**Technical Novelty And Significance:** 3
**Empirical Novelty And Significance:** 3
**Recommendation:** 6
**Confidence:** 3

**Main Review:**

Strengths
* This paper proposes a particular algorithm to attack vision transformers by considering the attention mechanism.
* This paper successfully identifies the vulnerability of ViT against dense patch attacks, while ViT was more robust under traditional Lp perturbations or natural perturbations in previous works.
* The paper has comprehensive experiments on benchmarking the robustness of ViT and CNN models, in terms of different patch selection strategy, number of perturbed patches, the sparsity of perturbation within each patch.

Weaknesses
* This paper doesn't consider the robust training for ViT to improve the robustness and evaluate the robustness of more robust models.
* I think "ARE VISION TRANSFORMERS ALWAYS ROBUST AGAINST ADVERSARIAL PERTURBATIONS?" in the title is not very meaningful, because it's known that it's impossible for a model to be robust against all perturbations so far, and even on the perturbations used in existing works, ViT is not robust. ViT was just relatively more robust than CNN.

**Summary Of The Paper:**

This paper studies the robustness of vision transformers (ViT) from the perspective of adversarial attacks on patches, where the attack algorithm is particularly designed to fool the attention mechanism. While some prior works show that ViT has better adversarial robustness compared to CNNs, this paper shows that ViT has worse robustness against patch attacks, when only a few patches are manipulated and the perturbations are dense within the patches.

**Summary Of The Review:**

This paper makes an important contribution on identifying a particular vulnerability of ViT against patch attacks compared to CNNs and has comprehensive empirical results, though this paper does not consider robust training for the models.

---

> ### Author Response · Authors · 2021-11-23
> **Response to Reviewer gMrG**
>
> Thanks for recognizing the importance of our work for identifying a particular vulnerability of ViT against patch attacks compared to CNNs. We have address your comments/concerns as follows and provided more experiments and analysis in the appendix of our updated manuscript.
>
> **1. Evaluating the robustness of ViTs with robust training**
>
> Thanks for your suggestion! We further benchmark the robustness of adversarially trained ViTs and CNNs, and still consistently observe the inferior robustness of ViTs against our proposed Patch-Fool attack. In particular, we apply Fast Adversarial Training (FAT) [1] with an $\epsilon$ of 2/255 and 4/255 under $L_{inf}$ constraint on top of DeiT-Ti and ResNet-18 on ImageNet. We report the robust accuracy of the FAT trained models against our Patch-Fool in the table below (“xP'' denotes the number of perturbed patches and "w/o" denotes the results without any adversarial training as reported in our submitted manuscript), from which we can observe that although FAT improves the robustness of both DeiT-Ti and ResNet-18 against our Patch-Fool attacks, DeiT-Ti is still more vulnerable against Patch-Fool than ResNet-18 under the same number of perturbed patches.
>
> |  |  |  | PGD-$L_{inf}$ | PGD-$L_{inf}$ | Patch-Fool | Patch-Fool | Patch-Fool | Patch-Fool |
> |:---:|:---:|:---:|:---:|:---:|:---:|:---:|:---:|:---:|
> | **Model** | **FAT** | **Clean Acc (%)** |  **$\epsilon$=0.2/255** |  **$\epsilon$=2/255** | **1P** | **2P** | **3P** | **4P** |
> | DeiT-Ti | w/o | 72.02 | 44.11 | 0.08 | 4.01 | 0.12 | 0 | 0 |
> |  | $\epsilon$=4/255 | 66.35 | 64.46 | 41.83 | 22.48 | 6.25 | 2.64 | 0.72 |
> | ResNet-18 | w/o | 71.78 | 30.49 | 0 | 12.66 | 2.60 | 0.10 | 0.12 |
> |  | $\epsilon$=4/255 | 60.13 | 58.57 | 37.18 | 23.48 | 11.58 | 6.09 | 2.69 |
>
>
> In addition, we further do more ablation studies on FAT trained DeiT-Ti against PGD attacks and our Patch-Fool as shown in the table below. We can see that (1) stronger adversarial training with larger $\epsilon$ leads to better robustness against both PGD attacks and our Patch-Fool, and (2) the improvement in robust accuracy against PGD attacks is higher than the one against Patch-Fool, indicating that enhanced adversarial training schemes or other defense methods are required to robustify ViTs against our Patch-Fool, which is also our future work.
>
> |  |  |  | PGD-$L_{inf}$ | PGD-$L_{inf}$ | Patch-Fool | Patch-Fool | Patch-Fool | Patch-Fool |
> |:---:|:---:|:---:|:---:|:---:|:---:|:---:|:---:|:---:|
> | **Model** | **FAT** | **Clean Acc (%)** |  **$\epsilon$=0.2/255** | **$\epsilon$=2/255** | **1P** | **2P** | **3P** | **4P** |
> | DeiT-Ti | w/o | 72.02 | 44.11 | 0.08 | 4.01 | 0.12 | 0 | 0 |
> |  | $\epsilon$=2/255 | 67.43 | 63.26 | 40.06 | 19.75 | 5.85 | 1.88 | 0.52 |
> |  | $\epsilon$=4/255 | 66.35 | 64.46 | 41.83 | 22.48 | 6.25 | 2.64 | 0.72 |
>
>
> We believe such discussions can further strengthen the contributions of our paper, and will include them in the final version and add more benchmarks on top of more adversarially trained models given more time.
>
> [1] “Fast is better than free: Revisiting adversarial training”, E. Wong et al., ICLR 2020.
>
>
> **2. Concerns for the title**
>
> Thanks for pointing this out! We adopt the current title to question the common belief that ViTs are more robust learners compared with CNNs as recognized by all previous works studying the robustness ranking between ViTs and CNNs (for more details, please check the third paragraph of the related work section). We hope this title can draw the community's attention to ViTs' vulnerability. Nevertheless, we totally agree that it's impossible for a model to be robust against all perturbations, and will think about and revise our title to explicitly specify the identified vulnerability of ViTs in the final version.

---

> > ### Comment · Reviewer_gMrG · 2021-11-29
> > **Post-rebuttal update**
> >
> > Thanks to the authors for the new experiments on adversarial training and the response, which are impressive to me. I would support acceptance. In addition, I’m hoping to see experimental details (e.g., hyperparameters) for adversarial training to be included in the next revision.

---

### Official Review · Reviewer_hTTd · 2021-11-03

**Correctness:** 4
**Technical Novelty And Significance:** 3
**Empirical Novelty And Significance:** 3
**Recommendation:** 8
**Confidence:** 4

**Details Of Ethics Concerns:**

To me there are no such concerns.

**Main Review:**

The strengths of this work include (a) extensive experiment to benchmark the robustness of different ViT varinst; (b) proposing a new attacj framework; (c) insightful findings.

Weaknesses: (a) Given those existing findings regarding the robustness comparison between ViT and CNN, this work might look a little incremental. This work clearly recognizes those works and I think it is not necessarily a negative point. (b) The authors mention that  their work might inspire innovative defense techniques. I do not see the rationale behind this claim. I suggest the authors remove this claim or illustrate it more clearly.

Minor suggestions and discussions:
A recent work [1] has also investigated the MLP-Mixer beyond ViT. It is suggested to discuss it. Is is possible to apply the proposed attack method also to MLP-Mixer. [1] also discusses universal attacks and I am curious whether the above attack method can be extended beyond Image-dependent attack to universal ones. Do the authors think the reasons that ViT are weaker against Patch-Fool might be explained from the shift invariance perspective [1,2]?


[1] Adversarial Robustness Comparison of Vision Transformer and MLP-Mixer to CNNs
[2] Shift Invariance Can Reduce Adversarial Robustness

**Summary Of The Paper:**

Given recent finding shows that ViTs are more robust than CNN, this paper investigates an intriguing question:
“Under what kinds of perturbations do ViTs become weaker learners compared to CNNs". They propose a Patch-Fool to fool the attention mechanism. Their investigation leads to some interesting findings and might inspire more interesting future work.

**Summary Of The Review:**

I believe this work is solid and provides insightful finding, and thus I vote for acceptance of this work.

---

> ### Author Response · Authors · 2021-11-23
> **Response to Reviewer hTTd: Part 1**
>
> Thanks for recognizing the insightful findings of our work. We have addressed your comments/concerns as follows and provided more experiments and analysis in the appendix of our updated manuscript.
>
> **1. Incremental contributions regarding the robustness comparison between ViT and CNN**
>
> We humbly emphasize that our work is the first to find that ViTs are more vulnerable than CNNs under patch-wise perturbations, which questions the common belief that ViTs are more robust learners compared with CNNs as recognized by all previous works (for more details, we refer the reviewer to the third paragraph of the related work section). Furthermore, we propose the notion of perturbation density to indicate the robustness ranking between ViTs and CNNs. As recognized by Reviewers gMrG and r33g, our work is “an important contribution on identifying a particular vulnerability of ViT against patch attacks compared to CNNs” and “is novel and the key finding of this paper is significant”. Finally, the benchmark in Section 3 of our submitted manuscript, covering various ViT/CNN variants and adversarial attacks, is more extensive than all previous works, which thoroughly validates the superior robustness of ViTs over CNNs under existing adversarial attacks.
>
> **2. Regarding the mentioned “might inspire innovative defense techniques”**
>
> Thanks for your suggestion! Our rationale is that the insights provided by our analysis about patch-wise perturbations can be utilized to guide the design of customized defense methods. For example, we have added the analysis about the patch-wise transferability of our Patch-Fool attacks in our updated manuscript in Appendix A.3, which shows experiment results when first generating the attack for one patch and then transferring the attack to other patch locations on the same image. From this set of experiments, we find that the transferability is relatively poor, especially between two patches far away from each other. Based on this observation, we suggest a simple defense strategy that randomly shifts the input image in a random direction by one or more patches, e.g., all patches will move towards one direction and the patches on the boundary will be either cropped or padded with zeros, thus there will be a mismatch between the patch for generating the attack and the one for applying the attack. To validate this, we gradually increase the number of random directions (from "only up and down" to all the four directions) and the maximal number of patches to be shifted (from 0 to 5, denoted as "shift range") as shown in the table below. We can observe that as we gradually increase the randomness, the robust accuracy will be notably increased while the clean accuracy can be maintained. Here "0" denotes no random shift with vanilla clean accuracy and robust accuracy against Patch-Fool.
>
> ||||||Shift Range||||
> |:---:|:---:|:---:|:---:|:---:|:---:|:---:|:---:|:---:|
> |**Model**|**Acc**|**0**|**1 (Only Up and Down)**|**1**|**2**|**3**|**4**|**5**|
> |DeiT-B|Clean Acc (%)|81.90|81.81|81.75|81.17|80.49|80.13|78.61|
> ||Robust Acc (%)|22.12|38.9|49.12|58.57|59.66|63.18|64.14|
> |DeiT-S|Clean Acc (%)|79.53|79.10|78.93|78.21|77.88|76.92|76.24|
> ||Robust Acc (%)|6.25|24.92|35.78|46.71|48.52|52.52|53.89|
> |DeiT-Ti|Clean Acc (%)|72.02|71.96|71.66|71.43|70.63|68.55|67.75|
> ||Robust Acc (%)|4.01|11.78|18.95|24.24|26.04|28.89|31.37|
>
> Note that such random transformation based defenses can be further weakened by adaptive attacks like EoT (Expectation over Transformation) [1] and here our key point is the insights provided by our work may shed light on new rounds of competitions between attack and defense techniques customized for ViTs. We will add more discussions in the final version.
>
> [1] “On Adaptive Attacks to Adversarial Example Defenses”, F. Tramer et al., NeurIPS 2020.

---

> > ### Author Response · Authors · 2021-11-23
> > **Response to Reviewer hTTd: Part 2**
> >
> > **3. Apply Patch-Fool on MLP-Mixer**
> >
> > Good question! Following your request, we apply Patch-Fool to MLP-Mixer B/16 with different numbers of perturbed patches as shown in the table below. We can see that (1) Patch-Fool can also notably degrade the accuracy of MLP-Mixer B/16 and (2) MLP-Mixer B/16’s robustness against Patch-Fool is better than DeiT-B and ResNet-152. We assume this is because the token-mixing MLP in MLP-Mixer shuffles the information between different tokens more aggressively than the self-attention mechanism in ViTs such that the original input patch of MLP-Mixer will quickly mix the information from other patches, leading to the reduced effectiveness of attacking its input patches.
> >
> > |Model|FLOPs|Clean Acc(%)|1 Patch|2 Patches|3 Patches|4 Patches|
> > |:---:|:---:|:---:|:---:|:---:|:---:|:---:|
> > |MLP-Mixer-B/16|11.6G|75.48|30.97|7.37|1.04|0.08|
> > |ResNet-152|11.56G|78.31|22.64|7.41|2.64|1.28|
> > |DeiT-B|17.58G|81.90|22.12|1.32|0.16|0.04|
> >
> >
> > **4. Extend Patch-Fool to universal attacks**
> >
> > Following your suggestion, we adopt UAP[2] on top of Patch-Fool, dubbed UAP-Patch-Fool. Without loss of generality, we select the center patch to build universal adversarial patches and transfer among the center locations of different images. As shown in the table below, we can observe that (1) UAP-Patch-Fool can successfully degrade the accuracy of both DeiT-B and ResNet-152, and (2) DeiT-B is more robust compared with ResNet-152 under UAP-Patch-Fool. We think that the difficulty for generating universal attacks for DeiT-B is because the transferability of Patch-Fool between different images of DeiT-B is poorer than that of ResNet-152. For example, we transfer the attacks generated for the center patch to all other images in the same batch and find that the averaged robust accuracy for DeiT-B / ResNet-152 is 78.16% / 69.14%, respectively, verifying the poor image-wise transferability of Patch-Fool on top of DeiT.
> >
> > | Model | UAP-Patch-Fool |  Vanila Patch-Fool |
> > |:---:|:---:|:---:|
> > | DeiT-B | 54.77 | 22.12 |
> > | ResNet152 | 38.49 | 22.64 |
> >
> > [2] “Understanding Adversarial Examples From the Mutual Influence of Images and Perturbations”, C. Zhang et al., CVPR 2020.
> >
> >
> > **5. Reasons behind the vulnerability of ViTs from the shift invariance perspective**
> >
> > Good question! In our opinions, the shift invariance perspective partly aligns with our analysis about the perturbation density (i.e., the number of perturbed pixels per patch), which influences the robustness ranking between ViTs and CNNs. In particular, the shift invariance of CNNs indicates that perturbing more pixels distributed among the whole image can incur a larger degradation on the robustness of CNNs than that of ViTs, i.e.,  a low perturbation density incurs more degradation in the robustness of CNNs. Similarly, the worse shift invariance of ViTs against CNNs makes them more vulnerable to perturbations around the region of interest, i.e., a high perturbation density (like our Patch-Fool) in one region can lead to ViTs’ inferior robustness.

---

> > > ### Comment · Reviewer_hTTd · 2021-11-25
> > > **My concerns are addressed.**
> > >
> > > Thanks for the rebuttal. It well addressed my concerns, thus I increase the score (a middle score like 7 might be more appropriate, though). It is suggested that the content in the rebuttal can be added to the revised version to make the work more complete and increase its impact.

---

> > > > ### Author Response · Authors · 2021-11-25
> > > > **Response to Reviewer hTTd**
> > > >
> > > > Thank you for recognizing the impact of our work! We will definitely add the above discussions to the final version of our paper.

---

### Decision · Program_Chairs · 2022-01-20

**Decision:**

Accept (Poster)

**Comment:**

This paper provides an interesting study on the adversarial robustness comparisons between ViTs and CNNs, and successfully challenges the previous belief that ViTs are always more robust than CNNs on defending against adversarial attacks. Specifically, as revealed in this paper, when the attacker considers the attention mechanisms, the resulting patch attack can hurt ViTs more.

Overall, all the reviewers enjoy reading this paper and appreciate the comprehensive robustness comparisons between ViTs and CNNs. The reviewers were concerned about the missing experiments about adversarial training, vague statements about the inspiration for future defenses, visualization of adversarial examples, etc. All these concerns are well addressed during the discussion period, and all reviewers reach a consensus on accepting this paper.

The final version should include the experiments, visualizations, and clarifications provided in the rebuttal. In addition, please release the code as promised.